# Comparing Learned and Iterative Pressure Solvers for Fluid Simulation

Category: Research

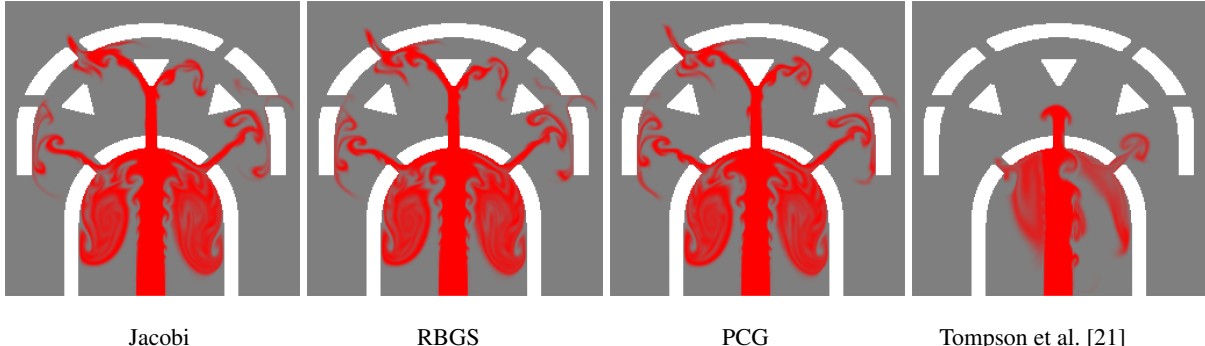

| Jacobi | RBGS | PCG | Tompson et al. [21] |

Figure 1: In the above *Channeling* scene at Frame $200$, a red plume is shot from the bottom centre. White regions represent obstacles, and each side of the scene is a wall (not shown explicitly). The set-up is identical across all four experiments, and we execute each iterative solver for $300$ ms. The fluid effect produced by each solver is shown accordingly. While different methods have varying plausibility, we ultimately evaluate the magnitude of the velocity divergence for different amounts of compute time in 2D fluid simulations such as this example.

## ABSTRACT

This paper compares the performance of the neural network based pressure projection approach of Tompson et al. [21] to traditional iterative solvers. Our investigation includes the Jacobi and preconditioned conjugate gradient solver comparison included in the previous work, as well as a red-black Gauss-Seidel method, all running with a GPU implementation. Our investigation focuses 2D fluid simulations and three scenarios that present boundary conditions and velocity sources of different complexity. We collect convergence of the velocity divergence norm as the error in these test simulations and use plots of the error distribution to make high level observations about the performance of iterative solvers in comparison to the fixed time cost of the neural network solution. Our results show that Jacobi provides the best *bang of the buck* with respect to minimizing error using a small fixed time budget.

**Index Terms:** Computing methodologies—Physical simulation; Computing methodologies—Neural networks;

## 1 INTRODUCTION

Simulating realistic fluid effects such as smoke, fire and water is an important problem in physically-based animation with broad applications in film and video games. In comparison to the accuracy required in computational physics and engineering domains, computer graphics research typically aims to provide fast and approximate techniques. The stable fluids work of Stam [18] is an excellent example of this, which has likewise had a strong influence on graphics research over the last two decades. The key idea is to take a splitting approach with the incompressible Navier-Stokes equations, and to use different integration methods for solving each term. A pressure projection to produce a divergence free velocity field is an important and costly step, and has been on its own a topic of research.

Our paper focuses specifically on comparing GPU implementations of iterative solvers with the work of Tompson et al. [21], which proposes the use of a learned convolutional neural network (CNN) to solve the pressure projection problem. With the growing popularity of learning techniques in physics simulation, we believe it is valuable to revisit this recent work using a set of tests that give us a better understanding of error and compute time in comparison to traditional

methods. In addition to the Jacobi and preconditioned conjugate gradient (solvers that were discussed by Tompson et al.), we also include experiments with red-black Gauss-Seidel to understand how the convergence properties change in comparison. We design three scenes to characterize typical 2D fluid simulation problems (e.g., see Figure 1 for one example), and collect statistics on the convergence across many time steps. With this data, we present the convergence in these simulations as distributions, allowing an understanding the typical behaviour of the different approaches.

## 2 RELATED WORK

Pressure projection is known to be one of the main bottlenecks for incompressible fluid simulation as it involves solving a Poisson equation (equivalently, a large-scale sparse linear system). The efficiency of pressure solver can often be the determining factor to overall speed performance of a fluid simulator. Such systems can be solved using iterative numerical methods including Jacobi, Gauss-Seidel (GS) and preconditioned conjugate gradient (PCG) [2, 7, 18, 19]. With the rise of multi-core CPUs and GPUs for general-purpose parallel computing, there has been a push to adapt classic numerical solvers in all variety of physics based simulation problems for these parallel hardware platforms [8, 14]. For instance, GS is inherently a sequential algorithm suitable only for running on a single thread. However, a small modification inspired by graph colouring leads to the red-black Gauss-Seidel (RBGS) method, which enables a parallel implementation which is useful for a variety of applications. For instance, Pall et al. [15] simulate cloth systems with projective dynamics solved with a GPU-based RBGS implementation, which performs more iterations than its CPU-based serial counterpart, but attains lower residual error given the same time budget.

Novel approaches for speeding up the pressure projection are important contributions in fluid simulation research. Extensions to multigrid approaches are one example that shows great promise in reducing computation times [8, 14]. McAdams et al. [14] employs multigrid as a preconditioner for their PCG solver. In contrast, Jung et al. [8] propose a heterogeneous CPU-GPU Poisson solver that does wavelet decomposition and smooth processing on CPU, and performs coarse level projection on GPU. In related work, Ando et al. [1] improve the coarse grid pressure solver initially presented by Lentine et al. [13], which reduces the size of the problem by

introducing a novel change of basis to help resolve free-surface boundary conditions in liquid simulation.

In contrast to numerical methods for fluid simulation, machine learning techniques present an interesting alternative. Yang et al. [23] present a data-driven method based on the neural network to infer pressure per grid cell that completely avoids the computations of traditional Poisson solvers. This approach significantly reduces the computational cost of pressure projection and makes it independent of the scene's resolution due to the algorithmic nature of forward propagation. The method still maintains a sufficiently good accuracy for visual effects compared to traditional Poisson solvers as long as the network is properly trained. However, their method does not generalize well to previously unseen scenarios. Tompson et al. [21] identifies test cases where the method of Yang et al. will fail, and propose a method to overcome these limitations. They instead pose the problem as an unsupervised learning task and introduce a novel architecture for pressure inference using a convolutional neural network (CNN), and their results show both greater stability and accuracy. A key part of their approach involves training and test sets representative of general fluids. They use wavelet turbulent noise [10] to produce random divergence-free velocity fields, and generate diverse boundaries a a collection of 3D models with varying scale, rotation and translation. They also employ data augmentation techniques, such as adding gravity, buoyancy, and vorticity confinement [20]. This helps their trained network be applicable to simulating fluids in a more general setting. In general, these learning based approaches exploit the observation that neural networks are very effective for regression, and are able to fit a deterministic function (a simulation problem) that maps inputs to the solution of an optimization problem.

Similar to the current trend for applications in other fields, machine learning has found increasing popularity for fluid simulations as a powerful tool in a general sense, not only for pressure projection. Ladický et al. [12] formulate the Lagrangian fluid simulation as a regression problem and use the regression forest method to predict positions and velocities of particles, possibly with a very large time step. With a GPU implementation, they obtain a $10\times$ to $1000\times$ speed-up compared to a state-of-the-art position-based fluid solver. Most recent work heavily adopted CNN-based methods to synthesize fluid flows thanks to its proven successful performance on computer vision problems such as image classification (e.g., AlexNet [11]). Chu et al. [4] use a CNN to learn small yet expressive feature descriptors from fluid densities and velocities. With the help of those descriptors, their simulation algorithm is capable of rapidly generating high-resolution realistic smoke flows with reusable space-time flow data. Kim et al. [9] trained a generative model with underlying CNN structure to reconstruct divergence-free fluid simulation velocities from a collection of reduced parameters. Their network includes an autoencoder architecture to encode parametrizable velocity fields into a latent space representation. They report a $1300\times$ increase in compression rates and a $700\times$ performance improvement compared to a CPU solver.

Beyond the core problem of fluid simulation, fluid super-resolution and controlling style is another direction of enormous interest that has seen great success in exploiting learning techniques. By leveraging the idea behind the generative adversarial network (GAN) [5]), Xie et al. [22] introduce tempoGAN, which comprises two discriminators, one being a novel temporal discriminator, to learn both spatial and temporal evolution of data for further synthesis of fluid flows with high-resolution details only from low-resolution data. We observe that learning techniques are broadly useful in fluid simulation, and in our work we put effort into better understanding their benefits and limitations within the core problem, specifically in the solution of the Poisson problem to produce divergence free velocity fields for incompressible fluids.

## 3 BACKGROUND

In this section, we give a brief review of the mathematical background and approaches to physically-based fluid simulation. We only focus on the two-dimensional case, but it is easy to generalize them to the three-dimensional space.

### 3.1 Fluid Equations

The motion of inviscid fluid flow is governed by the Euler equations, a special case of the famous Navier-Stokes equations [3],

$$\frac{\partial \mathbf{u}}{\partial t} + \mathbf{u} \cdot \nabla \mathbf{u} + \frac{1}{\rho} \nabla p = \mathbf{f} \qquad (1)$$

$$\nabla \cdot \mathbf{u} = 0 \qquad (2)$$

where Equations 1 and 2 are usually referred to as the momentum equation, which in fact is Newton's second law applied to fluids, and the incompressibility condition which restricts the volume of fluids to be constant, respectively. In these equations, $\mathbf{u}$ stands for velocity, $p$ is pressure, $\rho$ is density, and $\mathbf{f}$ is the total external force (e.g., gravity, buoyancy, vorticity confinement) acting on the fluid.

We spatially discretize the above PDEs on a marker-and-cell (MAC) grid, as done by Harlow and Welch [6], rather than a simple collocated grid to avoid the non-trivial nullspace problem, and to make the boundary condition easier to handle. Then, we use the finite difference method to approximate all partial derivatives. On the fluid-solid boundary, we require the normal component of the fluid velocity to be equal to that of the solid's velocity, that is,

$$\mathbf{u} \cdot \hat{\mathbf{n}} = \mathbf{u}_{\text{solid}} \cdot \hat{\mathbf{n}},$$

or their relative velocity has zero normal component.

### 3.2 Stable Fluids Framework

Suppose the current velocity field $\mathbf{u}^t$ at time $t$ is known, and we would like to advance our simulation to the next state and obtain $\mathbf{u}^{t+\Delta t}$ given time step $\Delta t$. It requires us to integrate the momentum equation while keeping the incompressibility condition satisfied, but the momentum equation involves three terms (not counting the velocity partial derivative with respect to time). Because it is challenging to handle all terms simultaneously, we use Stam's operator splitting approach to integrate them one by one [18]. This provides a modular framework, shown in Algorithm 1, which is first-order accurate (see Bridson's book [3]). Thus, we solve for the velocity field at time $t + \Delta t$ in three steps.

On line 3 of Algorithm 1, we first compute and add all external forces $\mathbf{f}$ over time step $\Delta t$ to the velocity field according to the standard forward Euler integration formula. Then on line 4, we self-advect the velocity field using the MacCormack method proposed by Selle et al. [17]. Finally, on line 5-6, we perform the pressure projection to make the resulting field $\mathbf{u}^{t+1}$ divergence-free by solving the Poisson problem

$$\nabla^2 p = \frac{\Delta t}{\rho} \nabla \cdot \mathbf{u}^B. \qquad (3)$$

---

1 Initialize a divergence-free velocity field $\mathbf{u}^0$;
2 **for** $t \leftarrow 0, 1, 2, \cdots$ **do**
3      $\mathbf{u}^A \leftarrow \mathbf{u}^t + \Delta t \mathbf{f}$;
4      $\mathbf{u}^B \leftarrow \texttt{Advect}(\mathbf{u}^A, \Delta t)$;
5      $p \leftarrow \texttt{SolvePoisson}(\mathbf{u}^B)$;
6      $\mathbf{u}^{t+1} \leftarrow \mathbf{u}^B - \frac{\Delta t}{\rho} \nabla p$;
7 **end**

**Algorithm 1:** Stable fluids solver.

This is equivalent to a linear system $\mathbf{A}\mathbf{x} = \mathbf{b}$ under our spatial discretization scheme, where $\mathbf{A}$ is a sparse, symmetric, and positive definite matrix often called five-point Laplacian matrix, $\mathbf{b}$ is a vector of velocity divergence of every fluid cell, and $\mathbf{x}$ consists of all pressure unknowns we desire to find. Stam [18] showed that solvers coupled with a semi-Lagragian advection algorithm such as Mac-Cormack and an accurate Poisson solver are unconditionally stable, meaning our simulation will not "blow up" no matter how large the time step $\Delta t$. As a result, we can arbitrarily vary the time step $\Delta t$ in accordance with our requirement to create fluid animations with different visual effects.

### 3.3 Iterative Methods for Linear Systems

Classic direct linear solvers, such as Cholesky factorization, can be used to solve the above Poisson equation if the number of unknowns is relatively small. When it comes to large-scale applications, however, those methods become impractical as they suffer from high time and space complexities. Even for fluid simulations where the matrix $\mathbf{A}$ is very sparse, the nonzero fill in the Cholesky factorization for large problems can become unattractive. Instead, we can use iterative linear solvers that gradually approximate the solution by repeatedly applying the update rule $\mathbf{x}^{k+1} = f\left(\mathbf{x}^k\right)$ specified by solver given an initial guess to the solution $\mathbf{x}^0$. Iterative methods can be fast compared to their direct counterparts, using minimal storage, and can be stopped at any time to yield an approximate solution. We briefly review the methods relevant to our experiments in this section, while Saad's book [16] provides an excellent reference for these techniques.

#### 3.3.1 Jacobi

The update rule of Jacobi method can be written as

$$\mathbf{x}_i^{k+1} = \frac{1}{\mathbf{A}_{ii}} \left( \mathbf{b}_i - \sum_{j \neq i} \mathbf{A}_{ij} \mathbf{x}_j^k \right), \tag{4}$$

where $\mathbf{A}_{ij}$, $\mathbf{x}_j^k$ denote the entry of matrix $\mathbf{A}$ at row $i$ and column $j$, and the $j^{\text{th}}$ component of vector $\mathbf{p}$ at time step $k$, respectively. Note that the positive definiteness of $\mathbf{A}$ implies $\mathbf{A}$ is invertible and all diagonal elements $\mathbf{A}_{ii}$ are positive, ensuring the validity of our update rule. Furthermore, every component of $\mathbf{x}^{k+1}$ depends on known values $\mathbf{x}_j^k$, hence Jacobi is inherently parallelizable, making it trivial to implement and efficient to run on GPU. However, this method tends to have a slow convergence rate and it may take many iterations before we get a satisfying solution.

#### 3.3.2 Gauss-Seidel (GS)

Consider modifying the update rule of Jacobi to use the updated values $\mathbf{x}_j^{k+1}$ once they are available. In this way, we obtain the Gauss-Seidel method defined as

$$\mathbf{x}_i^{k+1} = \frac{1}{\mathbf{A}_{ii}} \left( \mathbf{b}_i - \sum_{j<i} \mathbf{A}_{ij} \mathbf{x}_j^{k+1} - \sum_{j>i} \mathbf{A}_{ij} \mathbf{x}_j^k \right). \tag{5}$$

More precisely, for the $i^{\text{th}}$ component of $\mathbf{x}^{k+1}$, we compute the matrix-vector product by replacing $\mathbf{x}_j^k$ with $\mathbf{x}_j^{k+1}$ when $j < i$, and keep everything unchanged when $j > i$. Gauss-Seidel tends to exhibit a faster convergence rate, but loses Jacobi's parallel nature and becomes a sequential algorithm.

#### 3.3.3 Red-Black Gauss-Seidel (RBGS)

Graph colouring can provide a mechanism for obtaining the benefits of Gauss-Seidel while still allowing for parallelization. We can construct an undirected *dependency graph* $G$ as follows:

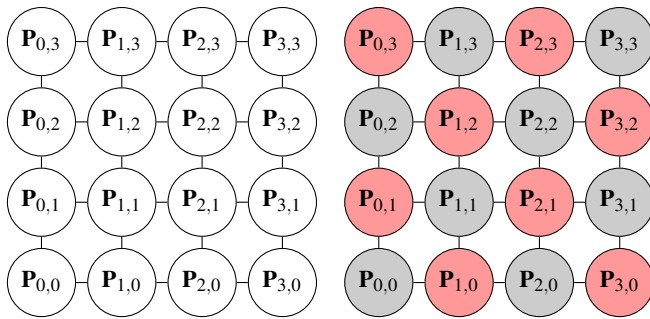

Figure 2: *Left*: The dependency graph for a $4 \times 4$ fluid pressure grid. *Right*: A red-black colouring partitions the graph into two independent sets. All vertices within the same set can be solved in parallel.

- Each component of $\mathbf{x}$ is a vertex;

- If the computation of $\mathbf{x}_i$ and $\mathbf{x}_j$ is interdependent, then add an edge $(i, j)$ to $G$.

At the $(i, j)$-entry of fluid grid, the Laplacian of the pressure $\mathbf{P}$ is equal to the divergence of the velocities $\mathbf{u}$. The discretized relationship is given by

$$4\mathbf{P}_{i,j} - \mathbf{P}_{i-1,j} - \mathbf{P}_{i+1,j} - \mathbf{P}_{i,j-1} - \mathbf{P}_{i,j+1} = (\nabla \cdot \mathbf{u})_{i,j}. \tag{6}$$

A vertex in graph $G$ corresponding to pressure $\mathbf{P}_{i,j}$ is connected to all its neighbours in a five-point stencil, namely, $\mathbf{P}_{i-1,j}, \mathbf{P}_{i+1,j}, \mathbf{P}_{i,j-1}$ and $\mathbf{P}_{i,j+1}$. We can colour this dependency graph by alternatively choosing from two colours, *red* and *black*. The coloured graph displays a checkerboard pattern, where a simplified example is shown in Figure 2. Observe that vertices of the same colour form an independent set, which indicates the computation of every vertex in this set can be done in parallel. As a consequence, we can solve Equation 3 in two parallel steps.

#### 3.3.4 Preconditioned Conjugate Gradient (PCG)

Given matrix $\mathbf{A}$ and vector $\mathbf{v}$, a Krylov subspace of $\mathbb{R}^n$ has the form

$$\mathcal{K}_n(\mathbf{A}, \mathbf{v}) = \text{span}\left\{ \mathbf{v}, \mathbf{A}\mathbf{v}, \mathbf{A}^2\mathbf{v}, \cdots, \mathbf{A}^{n-1}\mathbf{v} \right\}.$$

Conjugate Gradient (CG) is an orthogonal projection method onto the Krylov subspace $\mathcal{K}_n(\mathbf{A}, \mathbf{r}_0)$ where $\mathbf{r}_0 = \mathbf{b} - \mathbf{A}\mathbf{p}_0$ is the initial residual. It rephrases solving linear system as an optimization problem of the quadratic function

$$f(\mathbf{x}) = \frac{1}{2}\mathbf{x}^\mathsf{T}\mathbf{A}\mathbf{x} - \mathbf{x}^\mathsf{T}\mathbf{b},$$

in which the solution to the system $\mathbf{x}^*$ is its unique minimizer. CG then follows from the gradient descent algorithm. To improve convergence rate, we further incorporate an incomplete Cholesky preconditioner with zero-fill, IC(0). In our experiments, this is what we use for preconditioned conjugate gradients (PCG). CG and PCG expect the matrix $\mathbf{A}$ to be positive, symmetric, and definite, but eliminating variables by substituting boundary conditions into Equation 6 leads to a rank deficient matrix. To avoid the problems caused by matrix $\mathbf{A}$ being not full rank (both for the incomplete Cholesky preconditioner, and for the CG method), we instead solve the regularized system

$$(\mathbf{A} + \lambda\mathbf{I})\mathbf{x} = \mathbf{b}$$

for a carefully selected regularization parameter $\lambda$. Notice that in all our results we still report the residual norm as $r = \|\mathbf{b} - \mathbf{A}\mathbf{x}\|$ rather than $\|\mathbf{b} - (\mathbf{A} + \lambda\mathbf{I})\mathbf{x}\|$. Our regularized IC(0) PCG solver is given in Algorithm 2.

**Data: A, b**, regularization parameter $\lambda$
**Result: x**

1   $\mathbf{M} \leftarrow$ Generate-IC0-Preconditioner($\mathbf{A}$);
2   $\mathbf{A} \leftarrow \mathbf{A} + \lambda\mathbf{I}$;
3   $\mathbf{r}_0 \leftarrow \mathbf{b} - \mathbf{Ap}_0$;
4   $\mathbf{z}_0 \leftarrow \mathbf{M}^{-1}\mathbf{r}_0$;
5   $\mathbf{p}_0 \leftarrow \mathbf{z}_0$;
6   **for** $j \leftarrow 0, 1, 2, \cdots$ **do**
7      $\alpha_j \leftarrow \dfrac{\mathbf{r}_j^\mathsf{T}\mathbf{z}_j}{\mathbf{p}_j^\mathsf{T}\mathbf{Ap}_j}$;
8      $\mathbf{x}_{j+1} \leftarrow \mathbf{x}_j + \alpha_j\mathbf{p}_j$;
9      $\mathbf{r}_{j+1} \leftarrow \mathbf{r}_j - \alpha_j\mathbf{Ap}_j$;
10     **if** $\|\mathbf{r}_{j+1} + \lambda\mathbf{x}_{j+1}\| < \varepsilon$ *for given threshold $\varepsilon$* **then**
11       exit the loop;
12     **end**
13     $\mathbf{z}_{j+1} \leftarrow \mathbf{M}^{-1}\mathbf{r}_{j+1}$;
14     $\beta_j \leftarrow \dfrac{\mathbf{r}_{j+1}^\mathsf{T}\mathbf{z}_{j+1}}{\mathbf{r}_j^\mathsf{T}\mathbf{z}_j}$;
15     $\mathbf{p}_{j+1} \leftarrow \mathbf{z}_{j+1} + \beta_j\mathbf{p}_j$;
16   **end**

**Algorithm 2:** The regularized PCG solver used in our experiments.

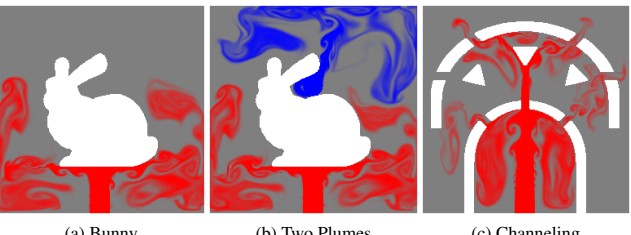

(a) Bunny      (b) Two Plumes      (c) Channeling

Figure 3: Three test scenes we designed and used to demonstrate and compare the performance of each pressure solver.

## 4   RESULTS

We use Jacobi, RBGS and PCG solvers implemented on the GPU in C++ with NVIDIA's CUDA, cuBLAS and cuSPARSE libraries. All experiments were carried out on a 3.6 GHz Intel i9-9900k CPU with 32 GB of RAM, and an NVIDIA GeForce RTX 2080 Ti GPU which contains 4352 shading units.

### 4.1   Test Scenes and Parameters

Figure 3 shows the three test scenes that we created to evaluate error and convergence of different solvers in different situations. The *Bunny* scene is designed to examine solver's basic performance in response to a simple scenario. It consists of an obstacle in the shape of a 2D Stanford bunny located at the centre of the domain, with an emitter shooting out red plumes from the midpoint of the bottom side. The *Two Plumes* scene is identical to the *Bunny* scene except that there is a second blue plume emitter placed in the gap behind the ears of bunny model. This scene mainly aims to test the case of multiple plume sources. Finally, we designed a *Channeling* scene, which has symmetry in vertical direction, to both test fluid channeling behaviours and to visually compare how well can different pressure solvers preserve the natural symmetry within the plume. In all scenes, white colours represent obstacles, and each side is a wall but not shown explicitly.

In order to obtain consistent and convincing results, we kept the test environment parameters identical across all our experiments. Specifically, we use a resolution of $256 \times 256$ pixels, and run every simulation for 300 frames with a step size of 0.25 seconds, and for the iterative solvers we executed 200 iterations for each. We do not use vorticity confinement, gravity, or buoyancy.

Notice that the CNN-based solver by Tompson et al. [21] is not an iterative method, so it does not have the concept of residual that converges like the Jacobi, RBGS, and PCG solvers. However, using the Helmholtz-Hodge decomposition we can observe that the residual is in fact equal to the norm of the divergence of the velocity at the next time step (see Appendix A). Thus we use this quantity as a means of comparing the iterative methods to the neural network approach.

### 4.2   Performance

Before examining and comparing performances of various methods, we need to first choose a suitable regularization parameter $\lambda$ for our PCG solver with respect to each scene. For the *Bunny* scene, we investigate the case with no regularization (or $\lambda = 0$) by running 400 PCG iterations and noted its relative residual, as shown in Figure 4(a). The simulation is executed for only one frame, because the relative residual diverges and animation of subsequent frames blows up.

Given the diverging residual that we observe with the non full rank matrix $\mathbf{A}$, we explore different amounts of regularization. Specifically we test how well $\lambda = 10^{-5}$, $\lambda = 10^{-4}$, $\lambda = 10^{-3}$ work, and repeat the same procedure as before. Figure 4(b)–(d) shows the order statistics (minimum, 25th percentile, median, 75th percentile, maximum) of relative residuals across all 300 frames at different iteration numbers. Since we only run 200 PCG iterations in our main comparisons, we choose $\lambda = 10^{-4}$ as it attains the lowest true-residual median at iteration 200 among all three choices of $\lambda$. Using the same criteria, we also selected $\lambda = 10^{-4}$ for the *Two Plumes* and *Channeling* scenes.

Figure 5 shows order statistics of relative residual over Jacobi, RBGS, and regularized PCG solver iterations for all $\mathbf{Ax} = \mathbf{b}$ systems and for each scene across 300 frames. We note that the iterations do not take the same amount of time, but the plots are still useful to observe the typical reduction in residual across different steps of the simulation.

We also show the order statistics of absolute residual $\|\nabla \cdot \mathbf{u}\|_2$ over total solving time (in milliseconds) for every scene in Figure 6. Since the forward propagation of CNN-based solver (Tompson et al. [21]) always takes a fixed amount of time, its performance is depicted as the intersection of horizontal and vertical purple lines in the graph. While we might expect PCG to have the fastest convergence rate and achieve lowest residual, it takes a longer time to solve compared to the highly parallel GPU implementations of the Jacobi, RBGS, and CNN-based solvers.

### 4.3   Robustness

An important feature of good numerical method is robustness, which we investigate by observing the behaviour of the four solvers when there is a sudden and fierce change of external conditions. In our stress-test, we monitor the $L_2$ norm of the velocity divergence of the pressure solvers to see how rapidly the residual returns to normal levels after facing a large external pressure is injected into the fluid region.

Figure 7 shows the set-up and result of our experiment. Our fluid simulator supports user interaction with a mouse to perturb the plume behaviour and it is with this mechanism that we inject a scripted mouse movement. We first run the simulation with the Jacobi solver for 100 frames, then at the moment that we inject the external pressure we switch to the different solvers to examine the post disturbance divergence norm trajectory.

The scripted mouse interactions used to inject a consist disturbance in each test, and the result of the disturbance, can be seen in Figure Figure 7(a)-(b). The disturbance lasts a total of 10 frames, with the scripted mouse drag first moving horizontally across the

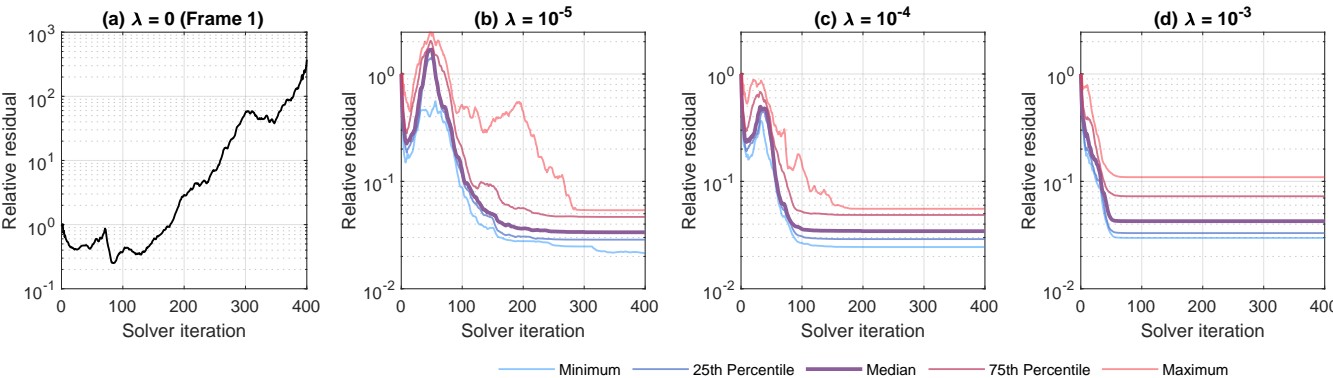

Figure 4: Convergence for the *Bunny* scene. (a) Without regularization, the relative residual over PCG solver's iteration for the $\mathbf{Ax} = \mathbf{b}$ system corresponding to frame 1 shows poor divergent behaviour. (b)–(d) Order statistics of the *true* relative residual at each PCG iteration (that is, measured without regularization), for all $\mathbf{Ax} = \mathbf{b}$ systems across 300 frames when the regularization parameter is $\lambda = 10^{-5}$, $\lambda = 10^{-4}$, $\lambda = 10^{-3}$.

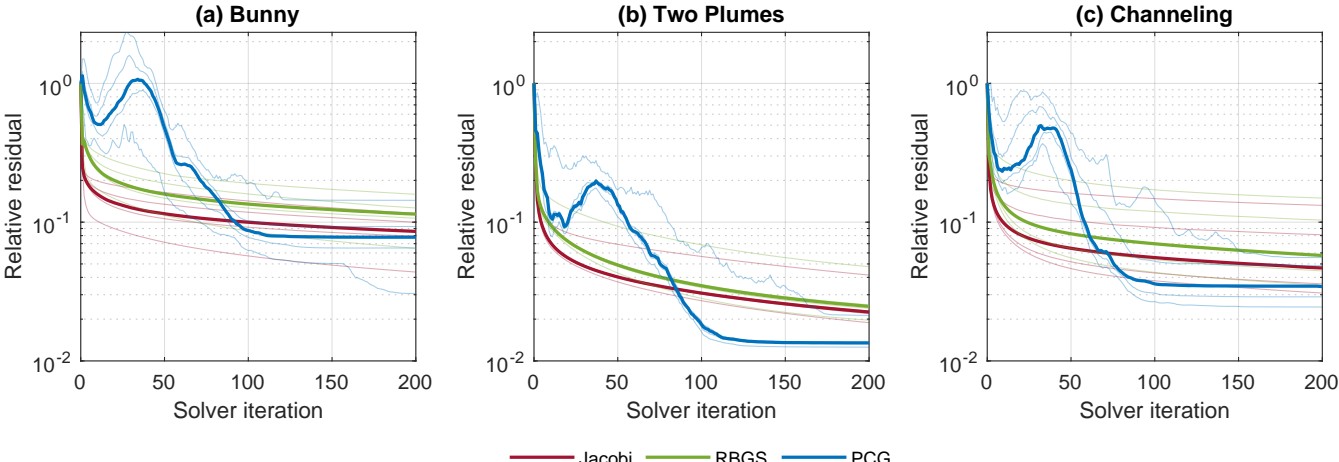

Figure 5: Order statistics of relative residual over each solver's iteration for all $\mathbf{Ax} = \mathbf{b}$ systems of each scene across 300 frames. Five curves from top to bottom of every solver represent maximum, 75th percentile, median, 25th percentile, minimum, respectively, and only curves for medians are in bold for clarity. PCG shows the true residual rather than that of the regularized problem being solved.

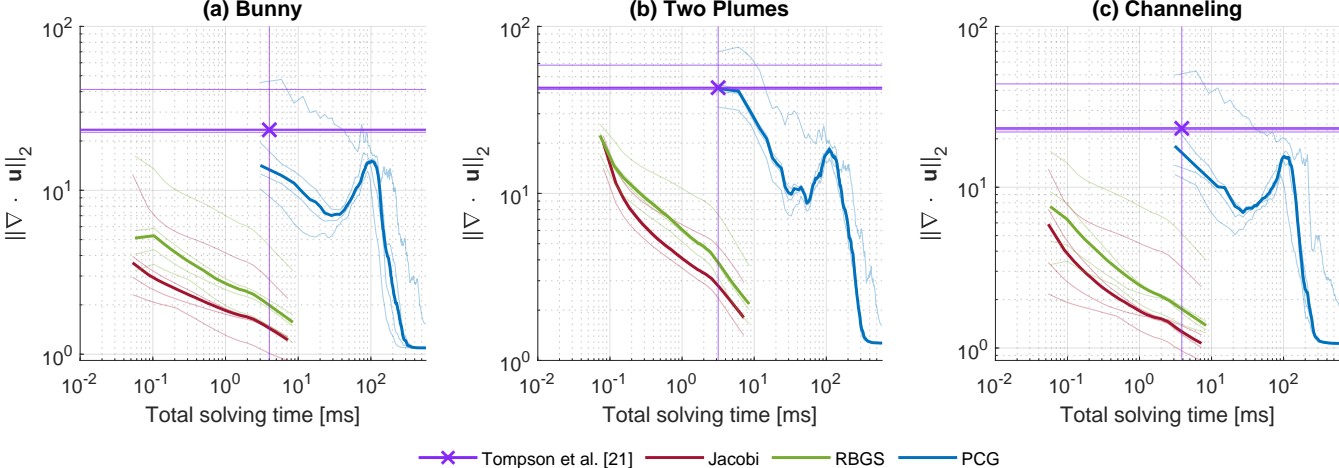

Figure 6: Order statistics of residual norm $\|\nabla \cdot \mathbf{u}\|_2$ over each solver's running time across 300 frames. Five curves from top to bottom of every solver represent maximum, 75th percentile, median, 25th percentile, minimum, respectively. Median curves are bold for clarity. Since the CNN-based solver (Tompson et al. [21]) always takes a fixed amount of time, its absolute residual over total solving time is shown as the purple intersection.

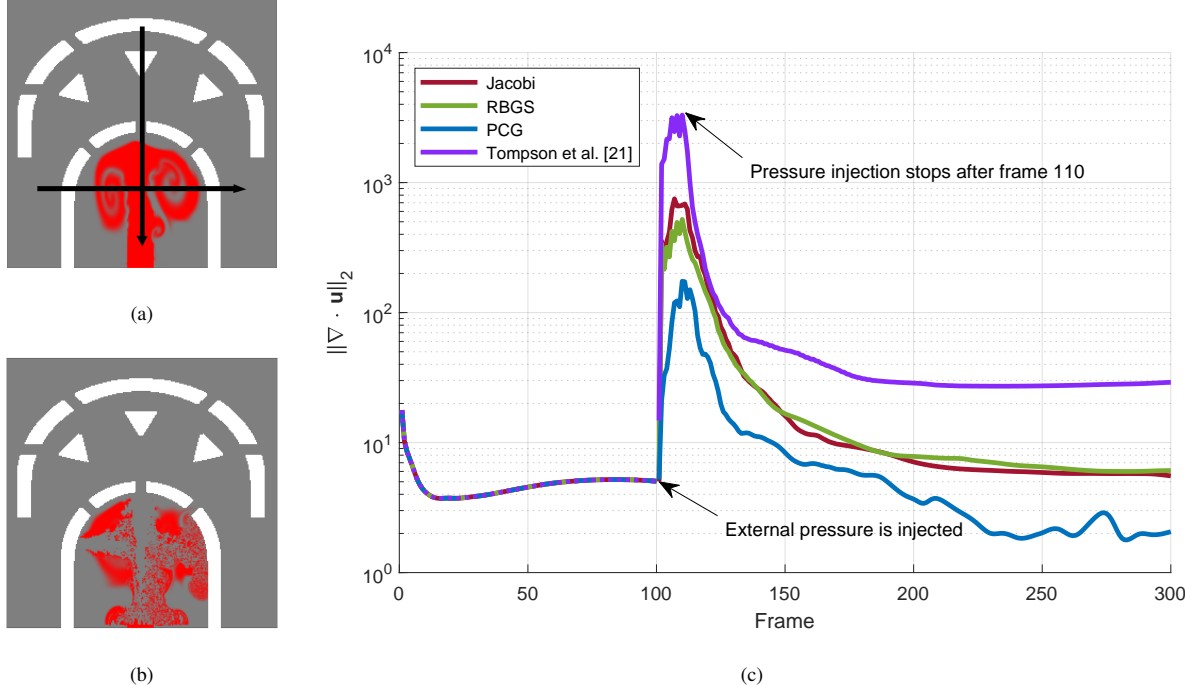

(a)

(b)

(c)

Figure 7: (a) The black arrow shows the mouse trajectory for external pressure injection. The mouse moves first horizontally for one frame, then vertically for the next frame, and this process is repeated 5 times for 10 frames violent scene perturbation. (b) Strong shape distortion of the red plume caused by our violent mouse perturbation. (c) $L_2$ norm of the velocity divergence over simulation frame count for different fluid solvers in our mouse perturbation experiment. The external pressure is injected at frame 101, and the injection terminates after frame 110.

scene for one frame, and then vertically in the next frame, with the process repeating 5 times. This produces a strong shape distortion of the red plume as seen in Figure 7(b). We then observe the norm of the divergence of the velocity after frame 110 to understand how high the norm rises, and then how quickly it drops to normal levels for the different solvers. The divergence norm trajectories in Figure 7(c) show that PCG has the most stable behaviour during the perturbation, while the method of Tompson et al. [21] fluctuates more violently than all others. After the perturbation ends, the norm of velocity divergence gradually drops to lower levels in all cases. Comparing the final divergence norm after the simulation stops at frame 300, we see that PCG achieves the lowest residual, followed by Jacobi and RBGS (with similar behaviour), while Tompson et al. [21] method has much higher residual and does not ultimately return to levels similar to those prior to the disturbance.

### 4.4 Discussion and Limitations

There are some high level observations we can make about our experiments. First, we expected RBGS to converge more quickly than Jacobi, at least with respect to the iteration count, but we did not observe this in our test scenes. We had also expected that PCG would have much better behaviour, it is perhaps still possible that with a different problem formulation and GPU implementation that we could see better advantages with PCG. We can note that there is likewise a vast number of other solvers we are not including in our comparison, such as those that exploit substructuring, or others based on multigrid. We believe all the results have good plausibility, but we also note that we were not able to observe preservation of symmetry in cases where it would be expected. That is, there is no guarantee that symmetry would be preserved with a neural network approach, and the arbitrary ordering of red before black in RBGS would have us not expect symmetry, but we had expected symmetric plumes in our Channeling scene when using Jacobi and

PCG. It is possible that our initial conditions and plume locations make it hard to achieve symmetry in these examples, and the single precision floating point used in the GPU implementations may yet be another explanation. Finally, it is interesting to note that the method of Tompson et al. actually *increases* the norm of the velocity divergence in our test scenes. While this is surprising, we note that the solutions it produces still tend to be useful for producing plausible incompressible flows.

### 5 CONCLUSION AND FUTURE WORK

We present a collection of tests to understand the convergence behaviour of different pressure solvers in 2D fluid simulations. Specifically, we compare Jacobi, red-black Gauss-Seidel, regularized preconditioned conjugate gradients, and a convolutional neural network (CNN) based solution proposed by Tompson et al. [21]. Our results show that Jacobi provides the best *bang of the buck* with respect to minimizing error using a small fixed time budget when running on the GPU. We note that the CNN solution has less desirable properties in our 2D tests, but may ultimately prove to be a good choice for specific scenarios, or larger 3D simulations. There are a number of interesting directions for future work, such as repeating experiments in 3D, and including comparisons with other solvers such as multigrid methods.

### A HELMHOLTZ-HODGE DECOMPOSITION

The Helmholtz-Hodge decomposition states that every vector field **u** can be decomposed into a divergence-free vector field **v** and a gradient field $\nabla q$, that is,

$$\mathbf{u} = \mathbf{v} + \nabla q.$$

Stam [18] shows that the divergence of this decomposition leads to the Poisson problem in Equation 3 needed for the pressure projection.

This problem has the form

$$\nabla^2 q = \nabla \cdot \mathbf{u}^B,$$

hence, the residual produced by iterative methods is defined as

$$\mathbf{r} = \nabla \cdot \mathbf{u}^B - \nabla^2 q.$$

Furthermore, the velocity update formula on line 6 in Algorithm 1 can be rewritten as

$$\mathbf{u}^{t+1} = \mathbf{u}^B - \nabla q.$$

Again, taking divergence of both sides of the above equation yields

$$\nabla \cdot \mathbf{u}^{t+1} = \nabla \cdot \mathbf{u}^B - \nabla^2 q$$
$$= \mathbf{r},$$

which justifies our comparison of the residual norm to the norm of the divergence of the next time velocity produced by the neural network approach.

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
