# OpenReview forum: "Comparing Learned and Iterative Pressure Solvers for Fluid Simulation"
_graphicsinterface.org/Graphics_Interface/2020/Conference — Submitted to GI 2020_

### Official Review · AnonReviewer2 · 2020-04-19

**Rating:** 6
**Confidence:** 3

**Review:**

This paper compares the performance of the learning-based pressure solvers of the Tompson et al. against the commonly used choices of interactive solvers for fluids simulation.

Although this paper is not proposing a new method, rather it compares the different approaches, it still gives some useful insight.  Hence, I think it is worth publishing paper.

The convergent graph shows similar convergence behavior for different problem settings (in Fig. 5 and in Fig 6). So the number of the setting is enough. The paper is more informative if it has 3D examples and comparisons between different resolutions.

The iterative solvers are implemented on GPU, not on CPU. This is reasonable because the approach by Topmoson et al. also use GPU. The performance of GPU solver is affected by its implementation so it would be nicer if the paper describes more how they are actually implemented using Nvidia's library.

The only downside of this paper is that it doesn't describe the learned-based solver well. The iterative solver described in Section 3 is fairly standard and doesn't require a lot of explanation. The paper should focus on illustrating how the learned-based solver was implemented (network architecture, training data, and training parameters).

Nevertheless, the paper did a nice job in comparison and it demystified the performance of the learned-based poison solver, I believe it worths publishing.

---

### Official Review · AnonReviewer1 · 2020-04-20

**Rating:** 3
**Confidence:** 4

**Review:**

Paper Summary: The paper compares some existing methods for solving the pressure projection step (Poisson problem) in a standard 2D fluid simulation, including a CNN-based approach, and concludes that the Jacobi method is preferable in terms of cost vs. error for a small fixed time budget.

Review:
From my perspective, lack of originality/significance is the main shortcoming of the paper. Its level of novelty is very low: it simply re-implements (or possibly just executes) a few existing techniques/codes and compares plots of their performance vs. error tradeoff. For the work to be broadly useful, it would need to compare a wider range of state-of-the-art alternative solver strategies, and preferably consider the three-dimensional case. There is simply not enough of a novel contribution in the current submission.

For example, the paper should have compared alternative CG preconditioners besides incomplete Cholesky, since IC is not very parallelizable, while using Jacobi or RBGS as a preconditioner to CG is more readily parallelizable. Likewise, solvers based on FFT and multigrid (e.g., "Low Viscosity Flow Simulations for Animation") should probably be considered. Similarly, since all the test domains are fixed in time and the problem is strictly two-dimensional and not huge, another useful point of comparison would be a direct solver approach, where the matrix A is prefactored once, so that only the forward/backward substitution phase needs to be executed at each step of the animation.

One potentially interesting insight from my perspective is that the CNN approach fares poorly compared to simpler traditional techniques, at least for the scenes considered. However, the authors don't really investigate/discuss why this (presumably) disagrees with the earlier findings of Tompson et al., who claimed that their method accelerates fluid simulation. Was it the GPU implementation? The focus on 2D? The time budget? Something else?

In addition, one important technical issue is that the use of regularization for the Poisson system is unnecessary. The null space in the system is a known one-dimensional space consisting of a constant offset of all the pressure values (since only the gradient matters to the subsequent velocity update). This null space can be eliminated directly by picking one active cell and assigning it a constant pressure value (e.g., zero). Therefore the regularization should be eliminated, and the experiments re-done. It is hard to know for certain a priori what effect this will have on the observed results.

Clarity: The paper is mostly clearly written aside from a few minor typos and grammatical errors. (By the way, the proper expression is "bang for the buck")  Section 3.3. can be removed and replaced with an appropriate reference to a linear algebra textbook, since it describes well-known existing linear solvers. This space might have been better spent reviewing Tompson's CNN-based scheme, since it is far less well-known/standard.

Additional comments:
The fact that RBGS converged more slowly than Jacobi even in terms of *iteration counts* is suspicious, since these are both standard techniques whose convergence behaviors are usually thought of as well-understood. Has the RBGS solver been verified on simpler model problems (e.g., Poisson on a square) to confirm that its implementation is indeed correct?

Did the authors simply reuse the existing code from Tompson et al.?

I would have liked to have seen a video comparing the visual results for the various solvers (unless I overlooked one in the submission?), since for a very small time-budget the remaining errors may yield quite different behavior, as seen in Figure 1. This is really mandatory for a paper in which the focus is computer animation.

---

### Official Review · AnonReviewer3 · 2020-04-24
**Not sure how representative the evaluation is**

**Rating:** 5
**Confidence:** 3

**Review:**

This is a well-written paper that benchmarks the performance of different pressure solvers for 2D fluid simulation, including the interesting case of learned (CNN) vs traditional solvers. I am not an expert in this area at all, so my comments are of a general nature. For the specific examples studied, the evaluation seems to be reasonably thorough and touches upon important aspects of interest. However, I do wonder how representative the set of test scenarios is. To my understanding, Tompson et al. also test against Jacobi and PCG, measure the divergence norm, and report conclusions that are (unsurprisingly) roughly opposite to those in this paper. Asuming both sets of authors are reporting their findings accurately, we can assume the difference boils down to specifics of the testing scenarios (2D vs 3D, initial and boundary conditions, hyperparameters, training data, etc). In which case, it is unclear what the practitioner should conclude from the current study, especially since it focuses only on the less realistic case of 2D flows. I think such studies are very valuable in general, but I am not sure whether the current one is broad enough.

Separately, this is obviously not a paper that proposes any new technical components, it is purely a benchmarking study of existing algorithms. I am not in a position to judge its suitability for GI on that score, I'll leave that to the meta-reviewer(s).

There are a few typos, e.g.
"The scripted mouse interactions used to inject a consist disturbance in each test, and the result of the disturbance, can be seen in Figure Figure 7(a)-(b)"

---

### Meta-Review · Area_Chair1 · 2020-04-24

**Recommendation:** Reject
**Confidence:** 5

**Metareview:**

Reviewer recommendations ranged from Clear Reject to Marginally Acceptable.

Pros:
-The paper performs comparisons of a few solvers for 2D fluid animation, including basic iterative solvers and a relatively recent learning-based method. Reviewers agreed that there can be value in benchmark studies of this kind.

Cons:
-The paper offers no novel simulation techniques.
-The range of solvers considered does not include state of the art techniques, e.g., multigrid, direct solvers (Cholesky), etc.
-Evaluation was restricted to 2D simulations.
-Minimal insight is offered as to why the findings disagree with those of Tompson et al.
-One reviewer raised technical concerns regarding unnecessary regularization and the observed convergence behavior of the author's RBGS implementation.

Since none of the reviewers advocated strongly for this work and several criticisms were raised, I am recommending rejection.

---

### Decision · Program_Chairs · 2020-04-25

Reject